# PD-L1 on dendritic cells attenuates T cell activation and regulates response to immune checkpoint blockade

Qi Peng[1,2], Xiangyan Qiu[3], Zihan Zhang[1], Silin Zhang[1], Yuanyuan Zhang[1], Yong Liang[3], Jingya Guo[4], Hua Peng [4], Mingyi Chen[3], Yang-Xin Fu [3] & Haidong Tang [1✉]

Immune checkpoint blockade therapies have shown clinical promise in a variety of cancers, but how tumor-infiltrating T cells are activated remains unclear. In this study, we explore the functions of PD-L1 on dendritic cells (DCs), which highly express PD-L1. We observe that PD-L1 on DC plays a critical role in limiting T cell responses. Type 1 conventional DCs are essential for PD-L1 blockade and they upregulate PD-L1 upon antigen uptake. Upregulation of PD-L1 on DC is mediated by type II interferon. While DCs are the major antigen presenting cells for cross-presenting tumor antigens to T cells, subsequent PD-L1 upregulation protects them from killing by cytotoxic T lymphocytes, yet dampens the antitumor responses. Blocking PD-L1 in established tumors promotes re-activation of tumor-infiltrating T cells for tumor control. Our study identifies a critical and dynamic role of PD-L1 on DC, which needs to be harnessed for better invigoration of antitumor immune responses.

---

[1] School of Pharmaceutical Sciences, Tsinghua University, 100084 Beijing, China. [2] Joint Graduate Program of Peking-Tsinghua-NIBS, School of Life Sciences, Tsinghua University, 100084 Beijing, China. [3] Department of Pathology, University of Texas Southwestern Medical Center, Dallas, TX 75235, USA. [4] Chinese Academy of Science Key Laboratory for Infection and Immunity, Institute of Biophysics, Chinese Academy of Sciences, 100101 Beijing, China. ✉email: hdtang@tsinghua.edu.cn

Antigen presentation is a dynamic event of an effective antitumor immune response[1]. During this process, antigen-presenting cells (APCs) capture and present tumor antigens to T cells, leading to their priming, activation, and possible reactivation[2]. Conventional dendritic cells (cDCs) are the most efficient APCs and consist two functional specialized subsets. Type 1 cDC (cDC1) cross-presents tumor antigens to CD8+ T cells[3,4]. By contrast, type 2 cDC (cDC2) is mainly associated with CD4+ helper T cell responses. Generally, it is believed that DCs play a central role in antitumor immune responses by mediating the priming and activation of T cells[5–7]. Interestingly, DCs might switch from immunostimulatory to immunosuppressive as tumors progress[8–10]. However, the mechanisms mediating such transition remain unclear.

PD-1/PD-L1 blockade therapy emerges as a promising new treatment for cancers in recent years[11–15]. Antibodies targeting PD-L1 are able to produce durable clinical responses in patients with a variety of cancers. In tumor microenvironment, PD-L1 is highly expressed in tumor cells as well as immune cells, such as DCs, macrophages, myeloid-derived suppressor cells (MDSCs), and Tregs[16]. However, the relative contributions of PD-L1 on these cells remain incompletely understood. PD-L1 on tumor cells is widely used as a biomarker for checkpoint blockade therapy. Assays measuring PD-L1 expression on tumor cells has been approved by US Food and Drug Administration as a companion diagnostic for PD-1/PD-L1 blockade therapy to determine whether a patient can benefit from the therapy[17]. However, almost a half of patients positive for tumor PD-L1 do not respond while some patients with PD-L1 negative on tumor cells can still respond to PD-L1 blockade, suggesting that the working model of how PD-1/PD-L1 signaling inhibits immune responses might be more complicated[18]. We and others have shown that, at least in some tumor models, PD-L1 on immune cells plays a more important role[19–27]. In the current study, we seek to further explore how PD-1/PD-L1 signaling works during host antitumor immune responses. We find that tumor-infiltrating DCs express a high level of PD-L1, which plays a critical role in limiting antitumor immune responses. In mouse model lacking PD-L1 on DCs, the therapeutic effects of PD-L1 blockade completely disappear. PD-L1 on DCs is upregulated during antigen-presentation to protect DCs from cytotoxicity of activated T cells. However, such effect also dampens the antitumor immune responses. These results provide additional insights into the mechanisms of immune checkpoint blockade therapy.

## Results

**PD-L1 on DCs is essential for checkpoint blockade therapy.** PD-L1 expression can be induced in many cells during inflammation responses. Our previous study has shown that blocking PD-L1 on myeloid cells is important for the responses to checkpoint blockade therapy[19]. To further identify which cell(s) is essential, we profiled PD-L1 expression levels on different cell types in tumor microenvironment by flow cytometry. PD-L1 expression on DCs was higher than other cells (Fig. 1a). In order to define the roles of PD-L1 on DC, we generated PD-L1-conditional knockout mice (Fig. 1b and Supplemental Fig. 1a). The mice were crossed with CD11c-cre to specifically knockout PD-L1 on DC. Flow cytometry data showed that PD-L1 expression was completely absent on DCs (CD11c+MHC II+) but not on other cells (Fig. 1c). Deficiency in PD-L1 does not affect DC development, as similar numbers of cells were found in the spleen and lymph node (LN) of conditional knockout mice (Supplementary Fig. 1b–f). To test the physical contribution of PD-L1 on DC, wild-type (WT) MC38 cells were inoculated into CD11c-cre;Pdl1fl/fl mice. Tumors grew slower in conditional knockout mice comparing to control mice (Fig. 1d).

Specifically, tumor sizes were ~600 mm³ in control mice at 29 days after inoculation, while the sizes were ~300 mm³ in DC-conditional PD-L1 knockout mice. There was no difference in PD-L1 expression by tumor cells (Fig. 1e). These data indicate a critical role of PD-L1 on DC for the antitumor immune responses. To measure the spontaneous immune responses against tumor, tissues were isolated from MC38 tumor-bearing mice and analyzed. T cell infiltration slightly increased in conditional knockout mice (Supplementary Fig. 2a). And there was a moderate increase of total CD8+ T cell activation in the absence of PD-L1 on DCs (Supplementary Fig. 2b, c). Next, we sought to evaluate antigen-specific responses. To measure endogenous antitumor immune responses, mice were challenged with OVA-expressing E.G7 cells. OT-1-specific T cells were enumerated by tetramer staining. More OT-1-specific CD8+ T cells were observed in DC-conditional knockout mice (Fig. 1f). To further characterize the functionality of DCs, mice were challenged with MC38 tumor expressing SIY as a model antigen. After tumor established, DCs were isolated from draining LNs (dLNs) and coincubated with naïve 2 C T cells. In the absence of PD-L1, DCs were more potent in priming T cells (Fig. 1g). These data suggest that PD-L1 on DCs plays important roles during T cell activation.

While most clinical trials focus on PD-L1 expression on tumor cells, cellular mechanisms by which PD-L1 suppresses cytotoxic T lymphocyte has not been well-defined due to the lack of confirmatory results. To evaluate the role of PD-L1 on DC for immunotherapy, we treated tumor-bearing conditional knockout mice with IgG or anti-PD-L1 antibody. Strikingly, MC38 tumors grew in DC-conditional PD-L1 knockout mice did not respond to PD-L1 blockade therapy at all (Fig. 2a). Another tumor model, E.G7, failed to respond to anti-PD-L1 as well (Supplementary Fig. 3a). A central role of DCs in T cell activation is their ability to present tumor antigens and to mediate T cell cross-priming[3]. Conventional DCs consist two functional different populations, cDC1 and cDC2. It has been reported that Batf3-deficient mice fail to generate cDC1s, which are important for antigen cross-presentation. Therefore, we challenged Batf3−/− mice with tumors and treated with anti-PD-L1. Although PD-L1 blockade therapy controlled tumor growth efficiently in WT mice, the same treatment failed to control tumor growth in Batf3−/− mice (Fig. 2b). Consistent with its profound role in CD8+ T cell priming, there was limited number of CD8+ T cell inside tumor microenvironment in Batf3−/− mice (Supplementary Fig. 3b). Collectively, these data suggest a key role of PD-L1 on DCs, likely cDC1s, in the responses to PD-L1 blockade therapy. In addition, DC-mediated antigen cross-presentation is essential for optimal tumor control by checkpoint blockade therapy.

**PD-L1 on DCs is upregulated by IFN-γ and T cells in tumor.** Since our data suggested that DC is critical for the responses to checkpoint blockade therapy, we measured PD-L1 on cDC1 and cDC2 by flow cytometry (Supplementary Fig. 3c). In naïve mice, cDC1 expressed a low level of PD-L1 in both spleen and LN (Fig. 3a, b). In MC38 tumor-bearing mice, cDC1 significantly upregulated PD-L1 in dLNs (Fig. 3b and Supplementary Fig. 3d). In addition, cDC1 in tumor microenvironment has the highest change in PD-L1 expression. By contrast, most cDC2 expressed a high level of PD-L1, which was further upregulated after tumor challenged (Fig. 3b). Similar results were observed in E.G7 model (Supplementary Fig. 3e). It is well-established that cDC can be generated ex vivo by bone marrow culture protocols with FLT3-L induction[28] (Supplementary Fig. 4). FLT3-L-induced bone marrow-derived DCs (BMDCs) expressed a low level of PD-L1 (Fig. 3c). These data suggest that PD-L1 on cDC1 is under dynamic regulations. PD-L1 expression can be upregulated by inflammatory cytokines, especially interferons (IFNs)[29]. Thus, we

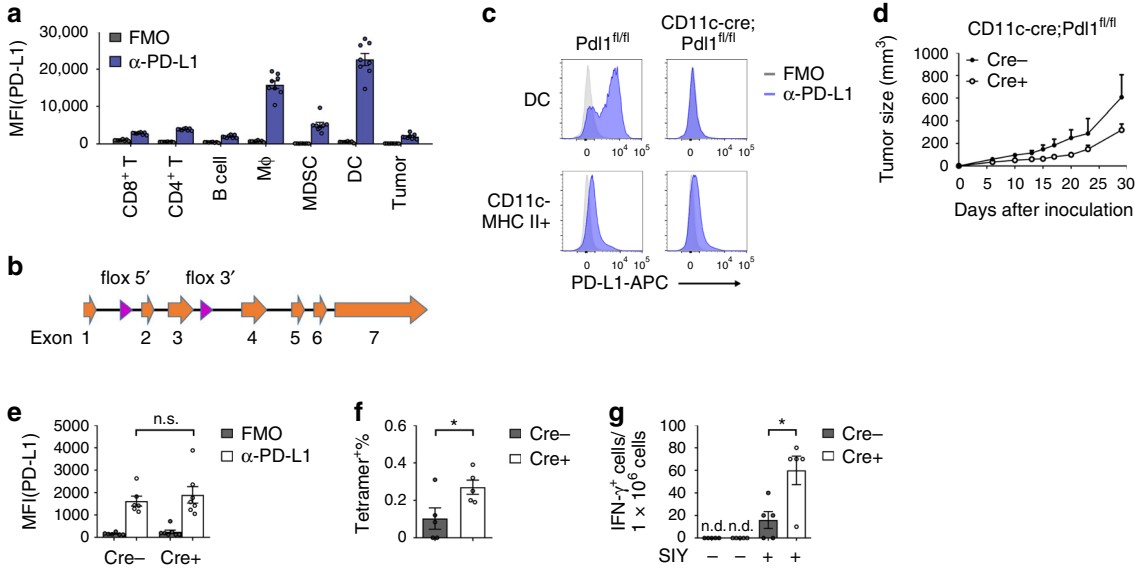

**Fig. 1 PD-L1 on DCs is important for T cell priming during antitumor immune responses. a** WT B6 mice ($n = 8$) were inoculated with $5 \times 10^5$ MC38 cells and analyzed by flow cytometry on day 14 after inoculation. PD-L1 expression levels on CD8+ T (CD3+CD8+), CD4+ T (CD3+CD4+), B (CD19+), macrophage (CD11b+F4/80+), MDSC (CD11b+Gr-1+), DC (CD11c+MHC II+), and tumor (CD45−) cells were shown. FMO, fluorescence minus one; MFI, mean fluorescent intensities. **b** Knockout strategy. LoxP sites were inserted flanking exons 2 and 3. **c** PD-L1 expression on DCs and control cells (CD11c−MHC II+) were measured by flow cytometry in CD11c-cre;$Pdl1^{fl/fl}$ and control mice. **d** CD11c-cre;$Pdl1^{fl/fl}$ or control mice ($n = 4$) were inoculated with $5 \times 10^5$ MC38 cells. Tumor sizes were measured twice a week. **e** MC38 tumor tissues were collected from CD11c-cre;$Pdl1^{fl/fl}$ or control mice ($n = 6$ Cre-, 7 Cre+) on day 14 after inoculation. PD-L1 levels on tumor cells were measured by flow cytometry. **f** CD11c-cre;$Pdl1^{fl/fl}$ or control mice ($n = 5$) were inoculated with E.G7 cells. Percentages of OVA-specific CD8+ T cells in PBMC were stained by tetramer on day 12. *$p = 0.0399$. **g** CD11c-cre;$Pdl1^{fl/fl}$ or control mice ($n = 5$) were inoculated with MC38-SIY cells. DCs were isolated from dLNs on day 14 and incubated with purified 2C T cells with or without SIY peptide restimulation. Forty-eight hours later, IFN-γ production was measured by ELISPOT. n.d., not detected. *$p = 0.0172$. Data are shown as mean ± SEM (**a**, **e**, **f**, and **g**) or mean + SEM (**d**) and are representative of two (**d**, **e**, **f**, and **g**) or three (**a** and **c**) independent experiments. n.s., not significant; *$p < 0.05$ determined by unpaired two-tailed Student's $t$ test. Source data are provided as a Source Data file.

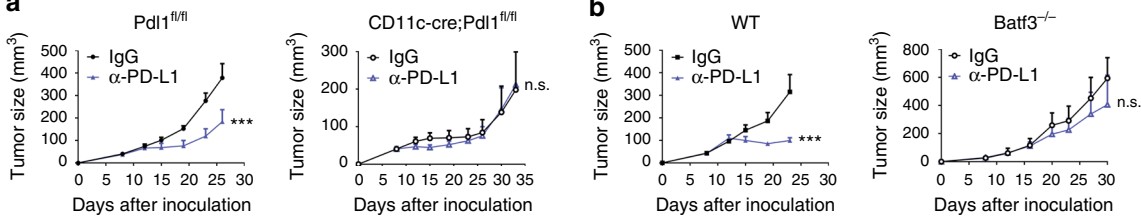

**Fig. 2 PD-L1 on DCs is essential for the response to PD-L1 blockade therapy. a** MC38 tumors established in CD11c-cre;$Pdl1^{fl/fl}$ ($n = 5$) or control mice ($n = 4$ IgG, 5 anti-PD-L1) were treated with IgG or anti-PD-L1 on day 8 and 12. Tumor growth curves were shown. ***$p = 0.0007$. **b** WT ($n = 4$ IgG, 5 anti-PD-L1) or $Batf3^{-/-}$ mice ($n = 3$ IgG, 4 anti-PD-L1) were inoculated with $5 \times 10^5$ MC38 cells. Mice were treated with IgG or anti-PD-L1 on day 8 and 12. Tumor growth curves were shown. ***$p < 0.0001$. Data are shown as mean + SEM and are representative of two independent experiments. n.s., not significant; ***$p < 0.001$ determined by two-way ANOVA. Source data are provided as a Source Data file.

investigate whether type I or II IFN signaling is involved in the upregulation of PD-L1 on DCs. In an in vitro system, treatment with both type I and II IFNs dramatically upregulated PD-L1 in BMDCs (Fig. 3d). Similar effects were observed in isolated primary cDCs (Fig. 3e). To evaluate the contributions of IFNs in vivo, type I or II IFN signaling was blocked by neutralizing antibodies. The upregulation of PD-L1 was significantly reduced in the absence of IFN-γ, while blocking type I IFN signaling had limited effects (Fig. 3f). Since CD8+ T cell is critical for antitumor immunity and is one of the major sources of IFN-γ, we wonder whether IFN-γ was produced by CD8+ T cells. As expected, depletion of CD8+ T cells reduced PD-L1 expression to a similar level as blocking IFN-γ (Fig. 3f). To recapitulate these effects in vitro, DCs were cocultured with activated T cells in the presence of neutralizing antibody against IFN-γ. PD-L1 upregulation significantly decreased in the absence of IFN-γ (Fig. 3g). In

summary, we found that PD-L1 on cDCs is upregulated by type II IFN and CD8+ T cells upon tumor challenge.

**PD-L1 is upregulated upon antigen uptake on type 1 DCs.** DCs play a central role for T cell priming. Specifically, cDC1 is the major APCs to carry tumor antigens from tumor tissues to draining LNs for T cell cross-priming[30]. To visualize antigen uptake in vivo, we inoculated mice with MC38-EGFP cells, which express EGFP as a reporter tumor antigen. Some cDC1s were positive for EGFP in tumor tissues and draining LN (Fig. 4a). By contrast, cDC2s took up antigens in tumor tissues while no/few EGFP-positive cDC2s were observed in dLN. To find out whether there is any relationship between PD-L1 expression and antigen presentation, we measured PD-L1 levels on DC subsets after antigen uptake. Intriguingly, EGFP-positive cDC1s showed the

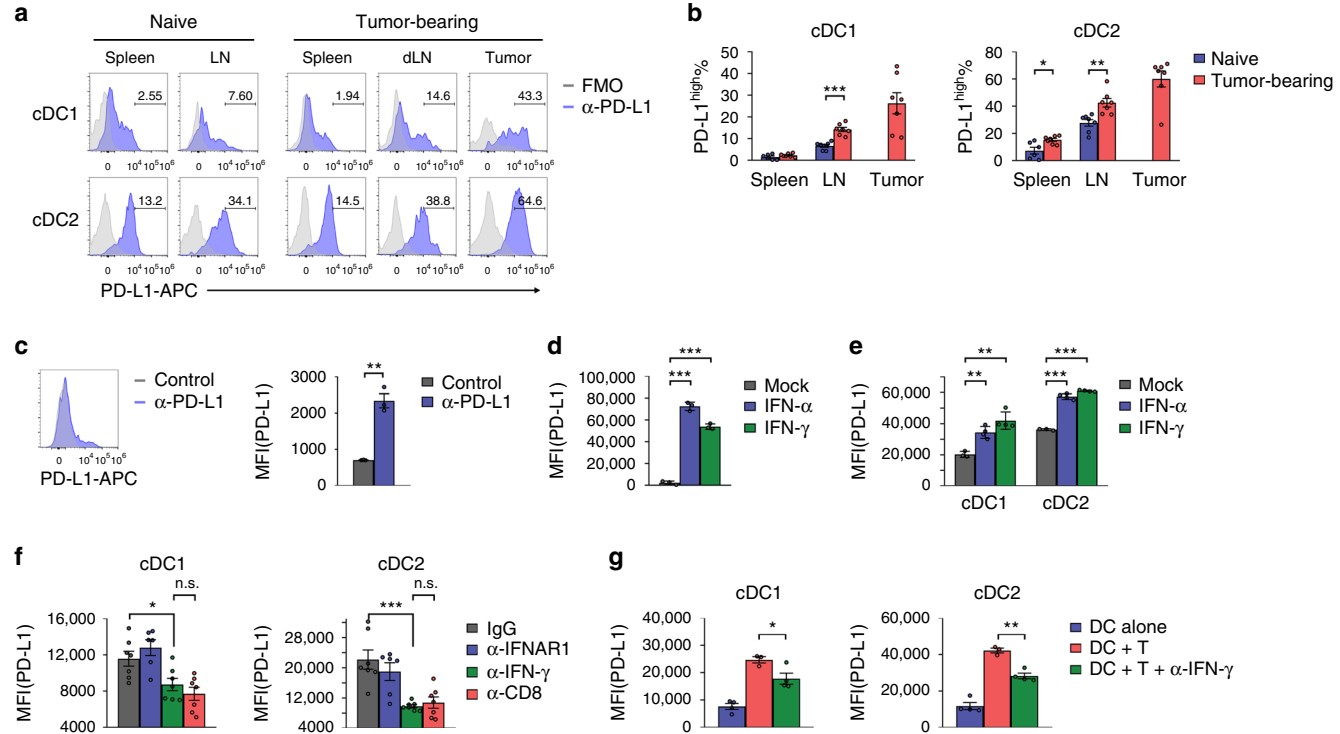

**Fig. 3 PD-L1 on DCs is upregulated by IFN-γ and T cells in tumor microenvironment. a, b** In naïve or MC38 tumor-bearing WT mice, PD-L1 levels on cDC1 (CD11b$^-$CD24$^+$) and cDC2 (CD11b$^+$CD24$^-$) were measured by flow cytometry ($n = 7$, except naïve spleen $n = 6$). ***$p < 0.0001$; *$p = 0.0106$; **$p = 0.0030$. **c** BMDCs were generated by FLT3-L. PD-L1 levels were measured by flow cytometry ($n = 3$). **$p = 0.0010$. **d** BMDCs were treated with recombinant IFN-α or IFN-γ. PD-L1 levels were measured 24 h later ($n = 3$). ***$p_{(IFN-\alpha)} < 0.0001$; ***$p_{(IFN-\gamma)} < 0.0001$. **e** Purified WT DCs were treated with IFN-α or IFN-γ. PD-L1 levels were measured 24 h later ($n = 3$ mock, 4 IFN-α, 4 IFN-γ). **$p_{(cDC1:IFN-\gamma)} = 0.0022$; **$p_{(cDC1:IFN-\gamma)} = 0.0015$; ***$p_{(cDC2:IFN-\alpha)} < 0.0001$; ***$p_{(cDC2:IFN-\gamma)} < 0.0001$. **f** C57BL/6 mice ($n = 7$, except anti-IFNAR1 $n = 6$) were inoculated with $5 \times 10^5$ MC38-EGFP cells. Mice were treated with IgG, anti-IFNAR1, anti-IFN-γ, or anti-CD8. PD-L1 levels on cDC1 and cDC2 in tumor tissues were measured by flow cytometry on day 14 after inoculation. *$p = 0.0208$; ***$p = 0.0005$. **g** Purified WT DCs and activated CD8$^+$ T cells were cocultured with or without anti-IFN-γ for 24 h. PD-L1 levels on cDC1 and cDC2 were measured by flow cytometry ($n = 4$, except DC + T $n = 3$). *$p = 0.0442$; **$p = 0.0013$. Data are shown as mean ± SEM and are representative of two independent experiments **a**, **c–e**, and **g** or pool of two independent experiments (**b** and **f**). n.s., not significant; *$p < 0.05$; **$p < 0.01$; ***$p < 0.001$ determined by unpaired two-tailed Student's $t$ test. Source data are provided as a Source Data file.

highest level of PD-L1 expression in the draining LN (Fig. 4b). In tumor tissues, we found that EGFP-positive cDC1s showed much higher PD-L1 expression comparing to EGFP-negative cDC1s (Fig. 4c). No significant difference was observed in cDC2s, suggesting that mechanisms regulating PD-L1 expression in these cells might be different. Ex vivo generated BMDCs were able to uptake antigens (Fig. 4d). However, there was less difference in PD-L1 expression after antigen uptake (Fig. 4e). These data suggested that PD-L1 upregulation in DCs was mediated by stimulations from other cells/cytokines in vivo. To identify the key factors, we neutralized type I or type II IFNs by antibody. Interestingly, even though antibody blocking did not affect antigen uptake, PD-L1 on EGFP-positive cDC1s was significantly reduced in the absence of type II IFN (Fig. 4f, g). Depletion of CD8$^+$ T cells showed similar effects (Fig. 4g). By contrast, the reduction of PD-L1 levels was mild in EGFP-negative cDC1s. These data demonstrate that PD-L1 expression on cDC1s is upregulated upon antigen uptake by IFN-γ and T cells.

**PD-L1 blockade reactivates tumor-infiltrating T cells.** PD-L1 blockade can either work by enhancing naïve T cell priming in draining LN, or by reactivating dysfunctional T cells in tumor tissues. To investigate which mechanism(s) is more important, we utilized FTY720 to block T cell infiltration to tumor tissues. FTY720 is a small-molecule analog of sphingosine 1-phosphate (S1P).

FTY720 treatment induces the internalization and degradation of S1P receptor, thus prevents lymphocyte egress from the LNs[31]. PD-L1 blockade therapy did not work if T cell trafficking was blocked in an early phase, such as eight days after tumor inoculation (Fig. 5a). Interestingly, if FTY720 was applied in a later time point when the tumors were more established, blocking T cell infiltration had no significant effects on the therapeutic effects of anti-PD-L1 (Fig. 5b). To validate the effects of FTY720 treatment, we measured T cell levels in peripheral blood and tumor tissues. Majority of the T cells disappeared from the peripheral blood immediately after FTY720 treatment (Supplementary Fig. 5a). The number of tumor-infiltrating T cells reduced as well (Supplementary Fig. 5b). PD-L1 blockade therapy significantly increased the number of activated (IFN-γ$^+$) CD8$^+$ T cells inside tumor (Fig. 5c). When T cell infiltration was inhibited in an early phase, the number of IFN-γ$^+$ T cells dropped significantly. By contrast, there was no difference in IFN-γ$^+$ T cells if FTY720 was applied in a later time point (Fig. 5c). Taken together, these data suggest that antitumor effects mainly depend on newly activated T cells in the LN in early stage tumors. However, as the tumors progress, sufficient T cells stay inside tumor tissues but they become more dysfunctional. And antitumor effects more depend on the reactivation of T cells inside tumors.

Under normal conditions, expression of PD-L1 protects the host from autoimmune diseases by preventing nonspecific activation or killing of the cytotoxic T lymphocytes[32]. Thus, we

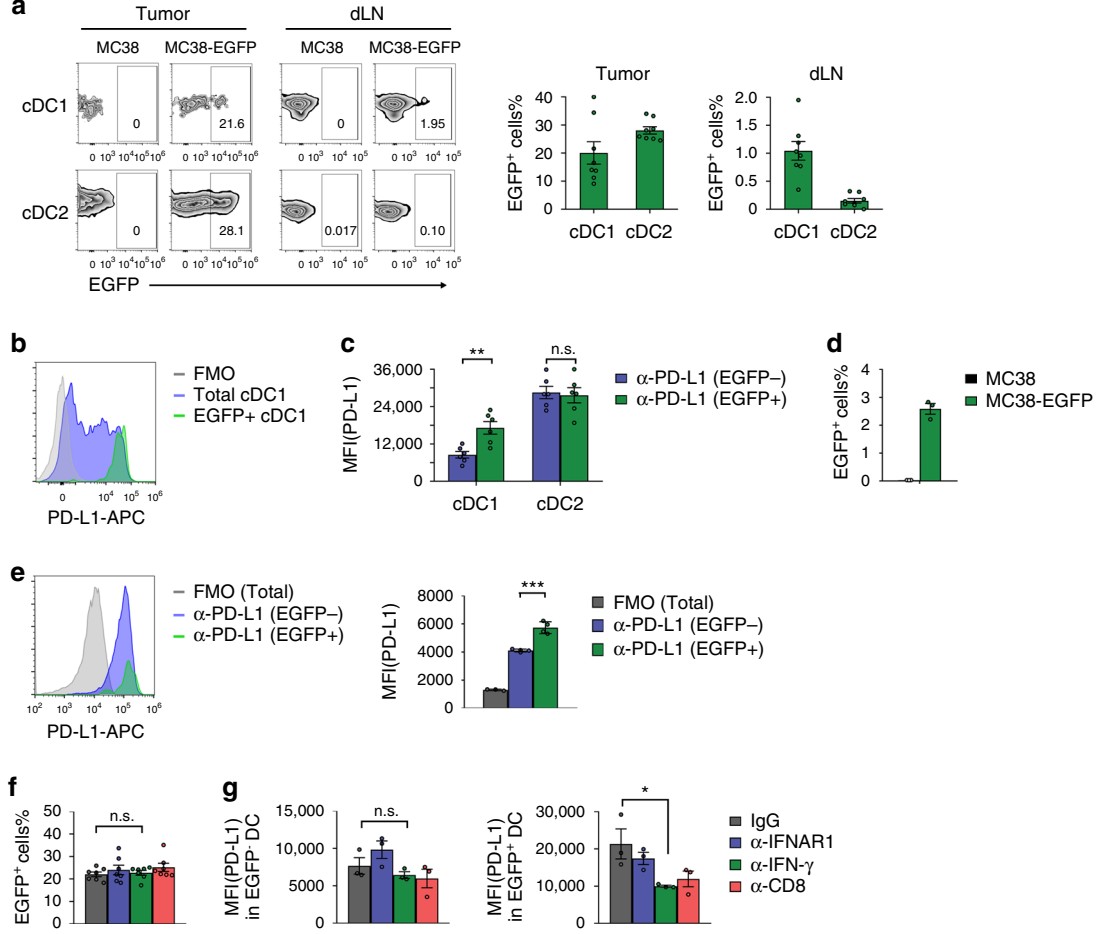

**Fig. 4 PD-L1 on cDC1s is upregulated during antigen presentation by IFN-γ. a** WT B6 mice were inoculated with $5 \times 10^5$ MC38-EGFP cells. After tumor established, tumors and draining LNs ($n = 8$ mice) were isolated on day 14. EGFP+ cells on cDC1 and cDC2 were measured by flow cytometry. **b** PD-L1 levels on total and EGFP+ cDC1s in draining LN were shown. **c** PD-L1 levels on EGFP− and EGFP+ DCs in tumor tissues were shown ($n = 6$ mice). **p = 0.0035. **d** BMDCs were co-culture with MC38 or MC38-EGFP cells for 24 h. Percentages of EGFP+ cells were shown ($n = 3$). **e** PD-L1 levels on EGFP− and EGFP+ BMDCs were shown ($n = 4$ cells, except FMO $n = 3$). ***p = 0.0003. **f, g** WT B6 mice were inoculated with $5 \times 10^5$ MC38-EGFP cells. Mice were treated with IgG, anti-IFNAR1, anti-IFN-γ, or anti-CD8 to block each pathway. Tumor tissues were collected and analyzed on day 14. **f** Percentages of EGFP+ cDC1s ($n = 7$ mice) were shown. **g** PD-L1 levels on EGFP− and EGFP+ cDC1s ($n = 3$ mice) were measured by flow cytometry. *p = 0.0489. Data are shown as mean ± SEM and are representative of two independent experiments (**b**, **c**–**e**, and **g**) or pool of two independent experiments (**a** and **f**). n.s., not significant; *p < 0.05; **p < 0.01 determined by unpaired two-tailed Student's t test. Source data are provided as a Source Data file.

wonder what the physical function of PD-L1 on DCs is. We measured DC survival by staining with live/dead dye. Interestingly, the survival of DCs was worse in the absence of PD-L1 (Fig. 5d). One of the functions of IFN-γ is to induce cytotoxicity. It has been proposed that PD-L1 signaling was able to protect tumor cells from such cytopathic effect[33]. We observed that IFN-γ was able to reduce viabilities of both tumor and dendritic cells (Fig. 5e). However, these effects could not be rescued by blocking PD-L1. As DCs closely contact T cells for antigen presentation, we wonder whether PD-L1 expression could protect them from cytotoxic killing by activated T cells. To recapitulate such killing in vitro, we isolated DCs from conditional knockout mice, loaded with OT-1 peptides, and coincubated them with activated OT-1 T cells. PD-L1-deficient cDC1s and cDC2s were more susceptible to killing by OT-1 T cells (Fig. 5f). The cytotoxicity was contact-dependent as it was abrogated in a transwell assay (Supplementary Fig. 6). Together, these data suggest that DCs upregulate PD-L1 expression to protect themselves from killing by cytotoxic T lymphocytes, which also dampers the antitumor immune responses. Blocking PD-L1 signaling on DC is essential for optimal antitumor immune responses.

## Discussion

The expression of PD-L1 is limited in normal condition. However, it can be upregulated during inflammation in many cells[34]. Due to the lack of proper animal models, the contributions of PD-L1 on cells other than tumor remains unclear[35]. Early study has shown that PD-L1 can be detected on patient-derived myeloid DC[36]. But its function in vivo is yet to be determined. In the current study, we sought to explore these questions by using cDC1-deficient mice and generating DC-conditional PD-L1 knockout mice. We observed that therapeutic effects of PD-L1 blockade therapy are completely disappeared in DC-conditional knockout mice, even though other cells still express high levels of PD-L1. Specifically, cDC1s, which are particularly important for the priming and activation of CD8+ T cells, upregulate PD-L1 expression upon antigen uptake. Consistent with a recent study, we found that PD-L1 on DCs protects them from killing by cytotoxic T lymphocytes, yet dampens the antitumor immune responses[37]. In addition, PD-L1 upregulation on cDC1s is mediated by IFN-γ produced by activated T cells. Such signaling forms a negative feedback to control T cell (re-)activation. Consistently, lack of cDC1s greatly reduces the number of

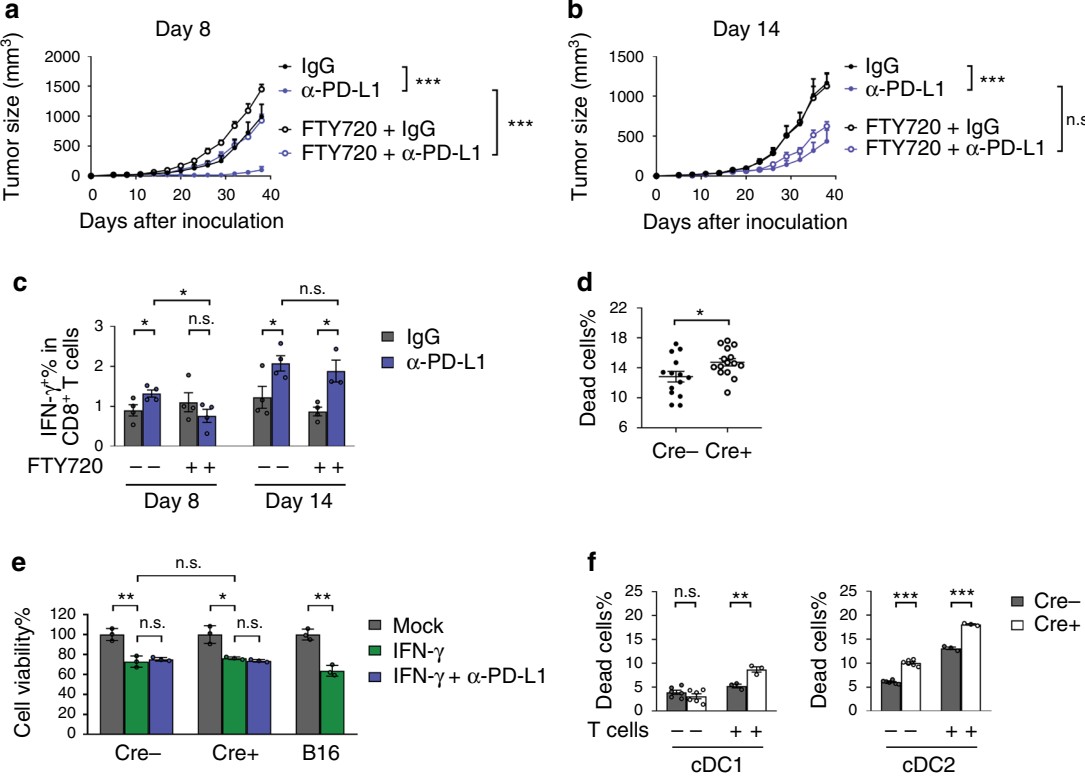

**Fig. 5 PD-L1 blockade reactivates T cells in tumor microenvironment for tumor control. a**, **b** C57BL/6 mice ($n = 6$, except day 14 FTY720 + anti-PD-L1 $n = 5$) were inoculated with $2.5 \times 10^5$ MC38-OVA cells and treated with (**a**) 200 µg IgG or anti-PD-L1 on days 8 and 12, or (**b**) 250 µg IgG or anti-PD-L1 on days 14 and 18. Mice were also treated with control or FTY720 the same day as antibody treatment. Tumor growth curves were shown. ***$p_{(8:\ IgG\ vs\ anti-PD-L1)} < 0.0001$; ***$p_{(8:\ anti-PD-L1\ vs\ FTY720+anti-PD-L1)} < 0.0001$; ***$p_{(14:\ IgG\ vs\ anti-PD-L1)} < 0.0001$. **c** Mice ($n = 4$, except day 14 FTY720 + anti-PD-L1 $n = 3$) were treated as in **5a** and **5b**. IFN-$\gamma^+$ CD8 T cells in tumor tissues were measured by flow cytometry on day 13 (for day 8 treatment group) or on day 19 (for day 14 treatment group). *$p_{(8:\ IgG\ vs\ anti-PD-L1)} = 0.0462$; *$p_{(14:\ IgG\ vs\ anti-PD-L1)} = 0.0443$; *$p_{(14:\ FTY720+IgG\ vs\ FTY720+anti-PD-L1)} = 0.0119$; *$p_{(8:\ anti-PD-L1\ vs\ FTY720+anti-PD-L1)} = 0.0245$. **d** CD11c-cre;$Pdl1^{fl/fl}$ or control mice ($n = 14$ Cre−, 15 Cre+) were inoculated with $5 \times 10^5$ MC38 cells. Viability of DCs was analyzed by flow cytometry on day 14. *$p = 0.0322$. **e** DCs purified from conditional knockout mice or B16 cells were treated with IFN-$\gamma$ for 96 h. Cell viability was determined by MTT assay ($n = 3$ cells). **$p_{(Cre-)} = 0.0048$; *$p = 0.0101$; **$p_{(B16)} = 0.0012$. **f** DCs isolated from spleens of CD11c-cre;$Pdl1^{fl/fl}$ or control mice were loaded with OT-1 peptide and incubated with activated OT-1 T cells at an E:T ratio of 3 for 4 h. Cell death of cDC1 and cDC2 cells were measured by flow cytometry ($n = 6$ -T cells, $3 +$ T cells). **$p = 0.0067$; ***$p_{(-T)} < 0.0001$; ***$p_{(+T)} < 0.0001$. Data are shown as mean + SEM (**a** and **b**) or mean ± SEM (**c**-**f**) and are representative of two independent experiments (**a**, **b**, **c**, **e**, and **f**) or pool of four independent experiments **d**. n.s., not significant; *$p < 0.05$; **$p < 0.01$; ***$p < 0.001$ determined by two-way ANOVA in (**a** and **b**) or by unpaired two-tailed Student's $t$ test in (**c**-**f**). Source data are provided as a Source Data file.

tumor-infiltrating lymphocytes. Therefore, PD-L1 on cDC1s is likely to prevent the overly expansion of tumor-infiltrating lymphocytes, and to protect the major APCs from killing by activated or reactivated T cells.

Besides PD-1, PD-L1 can also bind to another receptor CD80. An interesting fact is that APCs, including DCs, might express PD-1, PD-L1, and CD80 at the same time[38]. Furthermore, PD-L1 is able to interact with PD-1 and CD80 in cis, which makes the working model more complicated[38–42]. In an in vitro system, substantial CD80 expressed on APCs neutralizes PD-L1, thus preventing it from engaging PD-1 to inhibit T cell activation[40]. However, analysis of patient samples shows that PD-L1 is expressed much more abundantly than CD80 on DCs[43]. Therefore, it is likely that PD-L1 on DCs works to inhibit T cell activation in most scenarios. Indeed, blocking PD-L1 on human DCs augments their T cell priming ability[36,43]. However, which type(s) of DCs contributes to these effects has not been clear. Interestingly, by utilizing single-cell sequencing to analyze non-small-cell lung cancer samples, Maier et al. identified a subset of DCs which is enriched in immunoregulatory molecules[44]. Consistent with our data, they found that PD-L1 on DCs was upregulated after uptaking tumor antigens. However, the DC subset is found to be different from canonical cDCs[44]. These discrepancies might be

results of different cell/tumor models and experimental setups. In our current study, we utilized DC-conditional PD-L1-knockout mouse and tumor models responding to PD-L1 blockade therapy. We identified a critical and dynamical role of PD-L1 on DCs in T cell (re-)activation and immune checkpoint blockade therapy. The functional and mechanistic studies will provide new insights into the mechanisms of checkpoint blockade therapy.

During an effective antitumor immune response, APCs uptake and present tumor antigens to T cells, leading to T cell activation for tumor destruction[45]. In an established tumor microenvironment, majority of the infiltrating T cells are dysfunctional[46]. Whether PD-1/PD-L1 blockade therapy works by recruiting newly activated T cells or by reactivating tumor resident T cells is under debate[35,47]. Conclusions from clinical studies remain controversial[48]. It has been shown that majority of tumor-specific T cell clones after anti-PD-1 treatment are different from pre-existing tumor-infiltrating T cells in basal or squamous cell carcinoma[49]. Another study found that a subset of dysfunctional tumor-infiltrating T cells is able to respond to anti-PD-1 therapy efficiently in melanoma patients[50]. In animal models, it has been shown that T cell priming in LN is essential for checkpoint blockade therapy, as early surgical removal of LN after tumor inoculation eliminates the therapeutic effects[51,52]. Consistently,

we found that preventing newly activated T cells from entering tumor ablates majority of the antitumor effects in early stage tumors. However, inoculation of cultured cancer cells induces massive apoptosis and antigen release in the first few days. This process might result in artificial priming of T cells in the LN[53]. Therefore, we did the same treatment in late stage tumors, where most of these priming events were gone. We found that majority of the antitumor effects rely on tumor-infiltrating T cells. Our studies provide a potential explanation for the controversial observations in clinic. In these patients with massive tumor cell death (e.g., heavily treated patients) or strong immunogenicity, PD-1/PD-L1 blockade therapy may rely on both newly activated and reactivated T cells. By contrast, in patients with less antigen release or few functional T cells, efficacy of the therapy will more rely on reactivation of the pre-existing T cells. Nevertheless, our study highlights an important role of PD-L1 on DC for the inhibition of T cell (re-)activation, which needs to be harnessed for better invigoration of antitumor immune responses.

## Methods

**Mice**. Wild-type C57BL/6 mice were purchased from Vital River (Beijing, China). $Batf3^{-/-}$ and CD11c-cre mice were purchased from the Jackson Laboratory. $Pdl1^{fl/fl}$ mice were generated in the animal core. All mice were maintained under specific pathogen-free conditions at 22–26 °C with a 12:12 h dark/light cycle and 40–70% humidity. Wild-type female mice were used at an age of 6–8 weeks. For genetic-modified mice, age and sex matched mice were used for each experiment. Both female and male mice were used at an age of 6–12 weeks.

**Cell lines and reagents**. MC38, E.G7, B16F10, and 293 T cells were from American Type Culture Collection. Cells were cultured in 5% $CO_2$ and maintained in vitro in DMEM (MC38, B16F10, and 293 T) or RPMI-1640 (E.G7) medium supplemented with 10% fetal bovine serum (FBS), nonessential amino acids, 100 U/ml penicillin, and 100 μg/ml streptomycin. Recombinant murine IFN-α was produced as Fc-fusion protein in 293 T cells and purified by a protein A column.

**Generation of MC38-EGFP, MC38-OVA, and MC38-SIY cell lines**. In total, 293 T cells were transfected with three plasmids (pSIN-EGFP-puro, psPAX2, and pMD2. G) to produce lentivirus encoding EGFP. Forty-eight hours later, lentivirus was harvested and filtered through 0.45 μm filter. Wild-type MC38 cells were transduced with lentivirus. Two days after transduction, cells were treated with 5 μg/ml puromycin until resistant clones emerged. Single clones of EGFP-expressing MC38 cell (MC38-EGFP) were selected by limiting dilution. EGFP expression was confirmed by flow cytometry. MC38-OVA and MC38-SIY cell lines were generated similarly.

**Flow cytometric analysis**. Cells were blocked with anti-CD16/32 (clone 2.4G2) for 20 min at room temperature. Then cells were incubated with antibodies for 30 min. After washed, samples were acquired on a CytoFLEX S (Beckman Coulter) flow cytometer with CytExpert software (Beckman Coulter). Data were analyzed by FlowJo software (TreeStar).

**Tumor growth and treatments**. MC38 or E.G7 cells were subcutaneously injected into the right flank of the mice. When tumors reached certain size, mice were randomized to groups and treated intraperitoneally (i.p.) with IgG or anti-PD-L1 on indicated days. Tumor volumes were measured twice weekly and calculated as (Length × Width × Height/2). To inhibit lymphocyte trafficking, mice were treated i.v. with 50 μg FTY720 on day 8 or 14 after tumor inoculation. After first injection, ten micrograms FTY720 were given i.v. every day to maintain inhibition. For antibody blocking experiments, mice were treated with 200 μg IgG, 100 μg anti-IFNAR1, 200 μg anti-IFN-γ, or 200 μg anti-CD8 i.p. on day 0, 3, 7, and 11. Tissues were collected and analyzed on day 14.

**Cell isolation from tissues**. Tissues were cut into small pieces before digested in RPMI-1640 medium with 1 mg/ml type IV collagenase and 100 μg/ml DNase I. After digestion, tissues were passed through a 70 μm cell strainer to make single cell suspensions.

**T cell stimulation in vitro**. Splenic $CD8^+$ T cells were isolated with EasySep Mouse $CD8^+$ T cell Isolation Kit and stimulated with 5 μg/ml anti-CD3 (coated on plate) and 2 μg/ml anti-CD28 (soluble) according to standard protocol. Forty-eight hours after stimulation, T cells were harvested for functional assays. DCs were purified by FACS or EasySep Mouse CD11c Positive Selection Kit II according to the manufacturer's instruction. DCs and activated T cells were cocultured for 24 h with or without 10 μg/ ml anti-IFN-γ. PD-L1 levels on DCs were measured by flow cytometry.

**In vitro T cell killing assay**. Purified OT-1 T cells were activated by anti-CD3 and anti-CD28 for 48 h. Purified DCs were loaded with 5 μg/ml OT-1 peptides for 30 min at 37 °C. After washed, DCs were coincubated with activated OT-1 T cells for 4 h at an E:T ratio of 3. Cell death were measured by flow cytometry.

**Enzyme-linked immunosorbent spot assay (ELISPOT)**. Spleen or draining LN were isolated from MC38 tumor-bearing mice. Single cell suspensions were prepared. Cells ($4 \times 10^5$) were cultured for 24 h. In T cell priming assay, purified DCs ($1 \times 10^4$) from LN of MC38-SIY tumor-bearing mice were cocultured with naïve T cells ($1 \times 10^5$) purified from 2 C mice with or without 5 μg/ml SIY peptides for 48 h. ELISPOT assay was performed using an IFN-γ ELISPOT assay kit (BD Biosciences) according to manufacturer's instruction. Spots were enumerated by ImmunoSpot Analyzer (CTL).

**in vitro culture of BMDCs or primary DCs**. Bone marrow cells were obtained from WT mice. Cells were cultured at 10 cm Petri dishes in RPMI-1640 medium containing 10% FBS and 55 μmol/L 2-Mercaptoethanol. FLT3-L (150 ng/ml) were added every 3 days. BMDCs were harvested on day 9 and cocultured with irradiated MC38-EGFP cells at a ratio of 3:1 for 24 h. EGFP signal was measured by flow cytometry. In IFN-stimulation assays, BMDCs or purified DCs were treated with 500 ng/ml IFN-α or 950 ng/ml IFN-γ for 24 h. The concentrations of IFNs were chosen so that they induced similar levels of PD-L1 in MC38 cells. PD-L1 levels were measured by flow cytometry.

**MTT assay**. Cells were seeded in 96 well plates and treated with 10 ng/ml murine IFN-γ (and 10 μg/ml anti-PD-L1) for 96 h[33]. MTT assays were performed following manufacturer's instructions. Briefly, cells were incubated in 0.5 mg/ml MTT for 5 h at 37 °C. MTT crystals were dissolved in 150 μl DMSO and absorbance was measured in a plate reader at 570 nm. Cell viability was normalized to mock treated cells.

**Statistical analysis**. Data are shown as mean ± SEM. Differences between two groups were compared by an unpaired Student's two-tailed t test. Tumor growth curves were assessed by two-way ANOVA with Turkey's testing for multiple comparisons. A $p$ value < 0.05 was considered statistically significant. Statistical analysis was performed using the GraphPad Prism software (GraphPad). Tumor sizes were recorded and calculated by Excel (Microsoft). DNA gel images were acquired by Image Lab (Bio-Rad).

**Study approval**. Animal experiment protocols were consistent with guidelines of the Laboratory Animal Research Center of Tsinghua University. The laboratory animal facility has been accredited by Association for Assessment and Accreditation of Laboratory Animal Care International. All animal studies were approved by the Animal Care and Use Committee of Tsinghua University.

**Reporting summary**. Further information on research design is available in the Nature Research Reporting Summary linked to this article.

## Data availability

The data supporting the findings of this study are available within the article and its Supplementary Information files and from the corresponding authors on reasonable request. The source data underlying Figs. 1a, d–g, 2a, b, 3b–g, 4a, c–g, 5a–f, and Supplementary Figs. 1a–f, 2a–c, 3a, b, d, e, 5a, b, and 6 are provided as a Source Data file.

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

## Acknowledgements

We thank Dr. Wenxin Zheng for helpful scientific discussions. We thank services provided by the Laboratory Animal Research Center and Flow Cytometry Facility of Tsinghua University. This study was supported by Tsinghua-Peking University Center for Life Sciences (045-61020100119), Start-up Foundation of Tsinghua University (53332201119), and Foshan-Tsinghua Innovation Special Fund (FTISF, 041522068) to H.T.

## Author contributions

Q.P., X.Q., Z.Z., S.Z., and Y.Z. performed experiments. Q.P., X.Q., and H.T. analyzed data. Y.L., J.G., and H.P. provided reagents. Y.-X.F., X.Q., and M.C. contributed to manuscript preparation. H.T. designed experiments, wrote the manuscript, and supervised the project.

## Competing interests

The authors declare no competing interests.

## Additional information

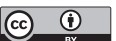

