## [Peer Review File · Nature Communications]

Reviewers' comments:

Reviewer #1 (Remarks to the Author):

This manuscript touches an important mechanism that is whether PD-L1 expression on dendritic cells plays a role in cancer immunology and immunotherapy. To address so, authors use a model in which floxed PD-L1 is depleted in all DC as a result of CD11c-cre selective expression. The subset of cDC1 cells is indirectly incriminated by experiments in BATF-3 KO mice and using cultures enriched in cDC1. Finally, authors make observations that support that the role of PD-L1 on DC is taking place in the tumor tissue based on FTY720 lymphocyte egression inhibition experiments. The information is substantially novel but a paper from Lieping Chen in PNAS (PMID: 29507197) made previous observations although that group focused on the fact that PD-L1 absence on DC resulted in more polyclonal T cell responses selectively attenuating the response of high avidity TCRs against model tumor antigens.

Comments:

1. The title is misleading since it refers to cDC1 but the paper does not focus the in-vivo effects of PD-L1 loss in this population. Another title could be suggested "PD-L1 on Dendritic cells attenuates T cell activation and is induced by Interferon gamma". This is because there is no convincing evidence for PD-L1 selectively on cDC1 and tumor escape is not actually observed. However, even if those points and claims are not experimentally met the paper still has value.
2. The magnitude of the effects in figure 1 D and E are rather modest. Evidence in additional tumor models aside of MC38 is clearly needed. Perhaps the effect would be more robust in other C57Bl6-syngeneic models such as EL4.
3. What about CD8 T cell infiltrates in CD11c-cre PDL1-flox/flox mice? Results in figure 1G are redundant with other publications and meaningless. This should be performed and shown with the PD-L1 KO mice if there is any difference. Results on BATF-3 mice could be deleted or presented as confirmatory supplementary information.
4. In line with the aforementioned paper by Lieping Chen the intensity of the immune response to defined MC38 tumor antigens should be monitored in WT mice versus mice with PD-L1 deficiency on their dendritic cells. MHC tetramers or in vitro T cell activation experiments should be undertaken.
5. In figure 2 it would be nice to see if PD-L1 upregulation is transcriptional and if type I interferon could result in a similar augmentation of surface PD-L1. Importantly it would be necessary compare if PD-L1 is induced by IFN γ on cDC2 as well since it is unlikely to be selective of the cDC1 subtype.
6. Figure 4 would benefit from experiments isolating CD11c+ DC from tumor and TDLN and studying if they stimulate T cells in culture and whether PD-L1 attenuates such an effect. For these purpose suitable TCR-transgenic T cells should be used.
7. PD-L1 has been proposed by David Escors group in a cell reports paper to mediate protective effects on bearing cells against IFN γ cytopathic effects (PMID: 28834746). Comparing DC from the CD11c selective PD-L1 KO and WT this point should be addressed.
8. Figure 4D is not convincing at all. The difference negligible and biologically irrelevant. Experiments in cocultures of CD8 cells and DCs expressing or not PD-L1 (using the PD-L1 selective KO mouse) should be undertaken.

Minor:

Editing by a native English speaker is recommended

Reviewer #2 (Remarks to the Author):

PD-L1 on type 1 dendritic cells promotes tumor immune escape by inhibiting T cell activation and re-activation

In this study the role of PD-L1 expression on type 1 conventional dendritic cells (cDC1) was investigated during immune checkpoint therapy in mouse models of colon cancer. The authors bred mice with a dendritic cell specific knockout of PD-L1 and treated these as well as mice lacking cDC1 cells and wildtype C57BL/6 mice with anti-PD-L1 antibodies after tumor inoculation. In these models the authors demonstrate that PD-L1 expressing cDC1 are essential in PD-L1 blockade therapy to reactivate tumor infiltrating lymphocytes to control tumor growth. Using Flow cytometry and ELISPOT authors claim that cDC1s upregulate PD-L1 expression upon antigen uptake mediated by T cell-derived IFN γ to avoid being killed by cytotoxic T lymphocytes. Despite being of clear clinical interest there are major issues that greatly limit the significance of this work.

Comments:

Even though the involvement of cDC1 in anti-PD-L1 therapy is novel and of potential importance to the immuno-oncology field, the scientific ground of these results is questionable. The current manuscript is limited in basic scientific aspects including textual clarity, number of references, number of replications of findings, number of experimental setups, completeness in reporting used materials and methods and profoundness in the discussion of findings. All of which should be expanded to make the article meet the scientific standard suited to the audience of the journal.

Specific comments:

- A large part of this article relies on bulk flow cytometry studies and therefore the gating strategy to identify cDC1 is crucial. In this study cDC1 is distinguished from cDC2 in the dendritic cell population (CD11c+MHCII+) using only CD24 and CD11b expression patterns. The authors need to show that their isolated cDC1 express specific markers such as XCR1, Clec9a, CD8 α or CD103.
- All key findings in this article are obtained using a singular tumor model: MC-38. Could these findings also be obtained in a separate tumor model? Secondly, anti-PD-L1 therapy is currently approved for the treatment of non-small cell lung cell carcinoma patients. Could the findings of this article also be reproduced in a clinically relevant tumor model for anti-PD-L1 therapy?
- In the majority of experiments fluorescently labeled antibodies are used. The authors should list all the fluorescently labelled antibodies used in this study and how these were obtained in the M&M section. Also authors made use of modified MC-38 cell lines including MC-38-eGFP and MC-38-OVA in their study that are currently not listed in the M&M section. The authors should make sure all elements necessary for interpretation and replication of the results are concisely written in the M&M section as stated in the Guide to Authors.
- Under statistical analysis authors state that all performed statistical tests were unpaired student's t tests. In Figures 2, 3, and 4 statistics were performed to compare the means of more than 2 groups and therefore correction for multiple testing should be applied. Please, authors should make sure appropriate statistics are applied in these experiments and report fully in the M&M section. Secondly, in all figures error bars are shown but authors should also state in figure legends what these error bars indicate.
- In Figure 1a, authors should mention which markers are used to define MDSCs. Could the authors also provide the gating strategies used to obtain the data depicted in this panel?
- In supplementary Fig 1 the CD11c-cre;Pd1fl/fl mice are characterized and authors claim PD-L1 deficiency does not affect DC development. Please add the frequencies of cDC1 and cDC2 in all relevant tissues. Also additional markers such as co-stimulatory receptors or MHC-class II should be measured to assess the state of cDC1 in knockout mice compared to wild type mice.
- In Sup. Fig 1ef authors look into the activational state of tissue resident T cells as possible mechanism for the essential role of PD-L1 expressing cDC1 in anti-PD-L1 therapy. More repetitions and additional experiments are needed to provide a proper understanding of PD-L1/PD-1 signalling between cDC1 and tissue-resident T cells leading to T cell activation.
- In Figure 2b, an explanation should be given for the assessment of PD-L1^{high} expressing cells (%) as opposed to total PD-L1 expression.
- In figure 2c more independent experiments plus quantification are needed of FTL-3 induced BMDC. Furthermore, the phenotype and vitality of generated cells should be provided to demonstrate robustness of BMDC cultures.
- In Figure 2 b+d, the expression level of PD-L1 is interrogated on cDC1. According the gating

strategy in Sup. Fig 2 cDC1 are distinguished from cDC2 in these experiments. Could the expression levels of PD-L1 on cDC2 also be made available to accompany current figures?

- In Figure 2d, additional experiments with isolated cDC1 cells should be performed to investigate the direct involvement of IFN γ and IFN α /IFN β on PD-L1 expression on cDC1.
- In Figure 3a authors study tumor antigen uptake by cDC1 isolated from tumor and dLN. More repetitions plus quantification are needed to draw conclusions. Secondly, authors claim that cDC1 take up antigens in the tumor and transport these to the dLNs. This is overstated because the current experimental setup does not allow drawing conclusions about the spatial and temporal dynamics of antigen uptake.
- In Figure 3a authors illustrate that 1.95% of cDC1 in the dLN have taken up tumor antigen and proceed to show a histogram of the PD-L1 expression in this small fraction of cells in Fig 1b. Could a representative dot plot including absolute cell count of gated cells be shown for this small portion?
- In Figure 3c, authors nicely show that PD-L1 expression is increased upon tumor antigen uptake in cDC1 but not in cDC2. Is it possible to provide representative FACS plots of MC-38eGFP uptake by cDC2 similar to cDC1 in Fig3a?
- Similarly in Fig 3d and e, an average of 2,5% of BMDCs take up tumor antigen of which the PD-L1 expression is shown. Here a representative and unadjusted dot plot should also be shown. Also more independent experiments plus quantification of PD-L1 expression on BMDCs are needed.
- From Fig3fg, authors conclude that PD-L1 expression on cDC1 is up regulated by antigen uptake, IFN γ and CD8+ T cells. Additional insights and direct evidence should be provided to explain the dynamics or mechanisms underlying these sources of PD-L1 regulation on cDC1.
- In Figure 4abc, FTY720 treatment is used to block infiltration by T cells into early or more established tumors. Please, provide data demonstrating the extent of migration inhibition by T cells in both early and late application of FTY720 treatment.
- In Fig 4d, authors claim that PD-L1 expression protects cDC1 from killing by cytotoxic T cells without any direct evidence. Additional experiments are required to support this conclusion.

We sincerely thank the reviewers for their valuable and constructive comments about our manuscript. We have addressed these comments with the addition of new data. We have provided a marked up version of the text, with major changes highlighted in yellow.

Here are the point-by-point responses:

Reviewer #1 (Remarks to the Author):

This manuscript touches an important mechanism that is whether PD-L1 expression on dendritic cells plays a role in cancer immunology an immunotherapy. To address so, authors use a model in which floxed PD-L1 is depleted in all DC as a result of CD11c-cre selective expression. The subset of cDC1 cells is indirectly incriminated by experiments in BATF-3 KO mice and using cultures enriched in cDC1. Finally, authors make observations that support that the role of PD-L1 on DC is taking place in the tumor tissue based on FTY720 lymphocyte egression inhibition experiments. The information is substantially novel but a paper from Lieping Chen in PNAS (PMID: 29507197) made previous observations although that group focused on the fact that PD-L1 absence on DC resulted in more polyclonal T cell responses selectively attenuating the response of high avidity TCRs against model tumor antigens.

Comments:

1. The title is misleading since it refers to cDC1 but the paper does not focus the in-vivo effects of PD-L1 loss in this population. Another title could be suggested

“PD-L1 on Dendritic cells attenuates T cell activation a and is induced by Interferon gamma”. This is because there is no convincing evidence for PD-L1 selectively on cDC1 and tumor escape is not actually observed. However, even if those points and claims are not experimentally met the paper still has value.

Response:

We thank reviewer for the comments and suggestions. We have revised the title as suggested.

2. The magnitude of the effects in figure 1 D and E are rather modest. Evidence in additional tumor models aside of MC38 is clearly needed. Perhaps the effect would be more robust in other C57Bl6-syngeneic models such as EL4.

Response:

Thank you for the suggestion. We repeated the experiment in E.G7 (a derivative of EL4) model (**Figure S3a** and **Figure S3d**). The results showed the same effects of PD-L1 on DC. Specifically, PD-L1-deficiency on DC abrogated the responses to PD-L1 blockade therapy in E.G7 model (**Figure S3a**). Comparing to naïve mice, cDCs upregulated PD-L1 in E.G7 tumor-bearing mice (**Figure S3d**). Together with the results from MC38 model, these findings strongly support our conclusions.

3. What about CD8 T cell infiltrates in CD11c-cre PDL1-flox/flox mice? Results in figure 1G are redundant with other publications and meaningless. This should be performed and shown with the PD-L1 KO mice if there is any difference. Results on BATF-3 mice could be deleted or presented as confirmatory supplementary information.

Response:

We measured the levels of CD8⁺ T cell infiltration in both PD-L1 cKO and KO mice (**Figure S2b**). In the spleen, there was no significant differences between Cre⁻, Cre⁺, or full KO mice. In the tumors, T cells slightly increased in Cre⁺ compared to Cre⁻ mice. The number further increased in full KO mice. Even though there was no significant difference between Cre⁺ and full KO, there was significant difference between Cre⁻ and full KO mice (p value determined by an unpaired Student's t-test). Results of Batf3^{-/-} mice were moved to supplement as suggested (**Figure S3b**).

4. In line with the aforementioned paper by Lieping Chen the intensity of the immune response to defined MC38 tumor antigens should be monitored in WT mice versus mice with PD-L1 deficiency on their dendritic cells. MHC tetramers or in vitro T cell activation experiments should be undertaken.

Response:

The intensity of immune response was evaluated with an MC38 tumor expressing SIY as a model antigen for better tracking early T cell response. In mice with PD-L1 deficiency on DCs, DCs were more potent in priming naïve 2C T cells, which specifically recognized SIY epitope (**Figure 1e**).

5. In figure 2 it would be nice to see if PD-L1 upregulation is transcriptional and if type I interferon could result in a similar augmentation of surface PD-L1. Importantly it would be necessary compare if PD-L1 is induced by IFN γ on cDC2 as well since it is unlikely to be selective of the cDC1 subtype.

Response:

These are great points! We measured PD-L1 mRNA level on DC by RT-qPCR (**Figure R1**). There was no significant differences in whole DC between naïve and tumor-bearing mice. However, PD-L1 was only upregulated in a small fraction of DCs (**Figure 2b**).

We found that both type I and II IFNs were able to upregulate PD-L1 on cDC1 and cDC2 in vitro (**Figure 2d**). However, only type II IFN was responsible for PD-L1 upregulation in vivo (**Figure 2e**).

Similar to cDC1, PD-L1 level on cDC2 reduced in the absence of IFNs in vivo (**Figure 2e**).

Figure R1. Relative PD-L1 mRNA expression in naïve and tumor-bearing mice. On day 14 after MC38 tumor inoculation, DCs were sorted from LN of tumor-bearing (n = 12) or naïve (n = 16) mice. Total RNA from DCs (1 µg) was subjected to RT-qPCR to measure the mRNA levels of PD-L1. Three different primers for PD-L1 were used. Expression levels were normalized to GAPDH. Each dot represents data from the pool of 4 mice (naïve) or 3 mice (tumor-bearing). Data are shown as mean ± SEM. n.s., not significant determined by unpaired Student's t-test. The following primer pairs were used: Pdl1-#1-F, GACCAGCTTTTGAAGGGAAATG, Pdl1-#1-R, CTGGTTGATTTTGCGGTATGG; Pdl1-#2-F, TGCTGCATAATCAGCTACGG, Pdl1-#2-R, GCTGGTCACATTGAGAAGCA; Pdl1-#3-F, CGTCACGATGGAGTGCAGAT, Pdl1-#3-R, AGGGCAGCATTTCCTTCAA; Gapdh-F, CATGTTCCAGTATGACTCCACTCACG, Gapdh-R, CCAGTAGACTCCACGACATACTCAGCA.

6. Figure 4 would benefit from experiments isolating CD11c⁺ DC from tumor and TDLN and studying if they stimulate T cells in culture and whether PD-L1 attenuates such an effect. For these purpose suitable TCR-transgenic T cells should be used.

Response:

As suggested, we isolated CD11c⁺ DC from dLN of MC38-SIY tumor-bearing mice, and co-cultured them with naïve 2C T cells (**Figure 1e**). PD-L1 attenuated the priming capacity of DC. We thank reviewer for the suggestions.

For tumor tissues, we were unable to isolate enough numbers of DCs for such assay due to low abundancy of DCs in tumor tissues and technical challenges, which is consistent with other studies (PMID: 30072305).

7. PD-L1 has been proposed by David Escors group in a cell reports paper to mediate protective effects on bearing cells against IFN γ cytopathic effects (PMID: 28834746). Comparing DC from the CD11c selective PD-L1 KO and WT this point should be addressed.

Response:

This is an interesting point! Consistent with David Escors' study, we observed that IFN- γ significantly inhibited the proliferation of B16F10 tumor cells (**Figure 4e**). The same treatment inhibited the proliferation of DC as well. However, we didn't observe any differences between

wild-type and PD-L1-deficient DCs. The effect was not affected by anti-PD-L1 antibody, either. These data suggest that such inhibition was independent of PD-L1 signaling in DCs.

8. Figure 4D is not convincing at all. The difference negligible and biologically irrelevant. Experiments in cocultures of CD8 cells and DCs expressing or not PD-L1 (using the PD-L1 selective KO mouse) should be undertaken.

Response:

To evaluate the protection effects of PD-L1 on DCs, we isolated DCs from conditional knockout mice and loaded them with OT-1 peptides. After that, DCs were co-cultured with activated OT-1 T cells. The viability of DC was measured by FACS (**Figure 4f**). In this system, PD-L1 expression significantly protected DC from T cell-mediated killing. These results are consistent with our previous data and further support our conclusions.

Minor:

Editing by a native English speaker is recommended

Response:

We have edited the language by professional editing.

Reviewer #2 (Remarks to the Author):

PD-L1 on type 1 dendritic cells promotes tumor immune escape by inhibiting T cell activation and re-activation

In this study the role of PD-L1 expression on type 1 conventional dendritic cells (cDC1) was investigated during immune checkpoint therapy in mouse models of colon cancer. The authors bred mice with a dendritic cell specific knockout of PD-L1 and treated these as well as mice lacking cDC1 cells and wildtype C57BL/6 mice with anti-PD-L1 antibodies after tumor inoculation. In these models the authors demonstrate that PD-L1 expressing cDC1 are essential in PD-L1 blockade therapy to reactivate tumor infiltrating lymphocytes to control tumor growth. Using Flow cytometry and ELISPOT authors claim that cDC1s upregulate PD-L1 expression upon antigen uptake mediated by T cell-derived IFN γ to avoid being killed by cytotoxic T lymphocytes. Despite being of clear clinical interest there are major issues that greatly limit the significance of this work.

Comments:

Even though the involvement of cDC1 in anti-PD-L1 therapy is novel and of potential importance to the immuno-oncology field, the scientific ground of these results is questionable. The current manuscript is limited in basic scientific aspects including textual clarity, number of references, number of replications of findings, number of experimental setups, completeness in reporting used materials and methods and profoundness in the discussion of findings. All of which should be expanded to make the article meet the scientific standard suited to the audience of the journal.

Response:

We thank reviewer for the comments and suggestions. We have revised the text and figures to address these helpful comments through the addition of new data and clarifications. Please find our responses below.

Specific comments:

- A large part of this article relies on bulk flow cytometry studies and therefore the gating strategy to identify cDC1 is crucial. In this study cDC1 is distinguished from cDC2 in the dendritic cell population (CD11c+MHCII+) using only CD24 and CD11b expression patterns. The authors need to show that their isolated cDC1 express specific markers such as XCR1, Clec9a, CD8 α or CD103.

Response:

We have further stained cDC1 and cDC2 for CD8a and CD103 (**Figure R2**). Splenic cDC1 (CD11b-CD24+) cells were positive for another marker CD8a, while cDC2 (CD11b+CD24-) were negative. A subset of cDCs were positive for CD103, representing the migratory DCs. These patterns are consistent with previous studies (PMID: 27637149).

Figure R2. Flow cytometry staining of DC subsets. Splenic cDC1 and cDC2 cells were further stained for CD8a and CD103. Data shown are representative of more than two independent experiments.

- All key findings in this article are obtained using a singular tumor model: MC-38. Could these findings also be obtained in a separate tumor model? Secondly, anti-PD-L1 therapy is currently approved for the treatment of non-small cell lung cell carcinoma patients. Could the findings of this article also be reproduced in a clinically relevant tumor model for anti-PD-L1 therapy?

Response:

These are great points! We have repeated our findings in another tumor model E.G7 (**Figure S3a and Figure S3d**). PD-L1 on DC was essential for antitumor responses in PD-L1 blockade therapy in E.G7 model (**Figure S3a**). PD-L1 on cDCs were upregulated in mice bearing E.G7 tumor (**Figure S3d**). These results were consistent with what we observed in MC38 model.

Specific for NSCLC, the most common used model in C57BL/6 background is Lewis Lung Carcinoma (LLC). However, several studies have shown that anti-PD-L1 therapy does not work in this model (PMID: 28096382, 29608143).

- In the majority of experiments fluorescently labeled antibodies are used. The authors should list all the fluorescently labelled antibodies used in this study and how these were obtained in the M&M section. Also authors made use of modified MC-38 cell lines including MC-38-eGFP and MC-38-OVA in their study that are currently not listed in the M&M section. The authors should make sure all elements necessary for interpretation and replication of the results are concisely written in the M&M section as stated in the Guide to Authors.

Response:

We are sorry for missing these details. We have provided information about the antibodies in **Supplementary Table 1**. Details about the MC38-EGFP and MC38-OVA cell lines were provided in the M&M section.

- Under statistical analysis authors state that all performed statistical tests were unpaired student's t tests. In Figures 2, 3, and 4 statistics were performed to compare the means of more than 2 groups and therefore correction for multiple testing should be applied. Please, authors should make sure appropriate statistics are applied in these experiments and report fully in the M&M section. Secondly, in all figures error bars are shown but authors should also state in figure legends what these error bars indicate.

Response:

We thank reviewer for the comments and we apologize for any confusions. We have provided details about the statistical analysis and error bars in each figure legend as well as in the M&M section.

- In Figure 1a, authors should mention which markers are used to define MDSCs. Could the authors also provide the gating strategies used to obtain the data depicted in this panel?

Response:

MDSC was defined as CD11b⁺Gr-1⁺. We have included it in the revised figure legend. The gating strategies for **Figure 1a** were shown (**Figure R3**).

Figure R3. Gating strategies for figure 1a. (a) Tumor tissues were collected as in **figure 1a**. Single live CD45⁺ cells were gated. (b-f) Single live CD45⁺ cells were further gated for (b) CD8⁺ T (CD3⁺CD8⁺) and CD4⁺ T (CD3⁺CD4⁺), (c) B (CD19⁺), (d) macrophage (CD11b⁺F4/80⁺), (e) MDSC (CD11b⁺Gr-1⁺), and (f) DC (CD11c⁺MHC II⁺) cells.

- In supplementary Fig 1 the CD11c-cre;Pd11fl/fl mice are characterized and authors claim PD-L1 deficiency does not affect DC development. Please add the frequencies of cDC1 and cDC2 in all relevant tissues. Also additional markers such as co-stimulatory receptors or MHC-class II should be measured to assess the state of cDC1 in knockout mice compared to wild type mice.

Response:

As suggested by the reviewer, the frequencies of cDC1 and cDC2 were measured (**Figure S1b-S1f**). When comparing naïve Cre-positive mice to Cre-negative littermates, there was a slight

increase of cDC1 (Cre-: 15.550±0.240% vs Cre+: 17.200±0.212%, p=0.0021) in spleen and decrease of cDC1 (Cre-: 32.050±0.641% vs Cre+: 28.050±1.015%, p=0.0158) in dLN (**Figure S1c**). The same parameter had no difference in MC38 tumor-bearing mice. There was no significant difference in total DC (**Figure S1b**), cDC2 (**Figure S1d**), MHC II levels in cDC1 (**Figure S1e**), and MHC II levels in cDC2 (**Figure S1f**). (Data are shown as mean ± SEM. Differences were compared by an unpaired Student's t test.)

- In Sup. Fig 1ef authors look into the activational state of tissue resident T cells as possible mechanism for the essential role of PD-L1 expressing cDC1 in anti-PD-L1 therapy. More repetitions and additional experiments are needed to provide a proper understanding of PD-L1/PD-1 signalling between cDC1 and tissue-resident T cells leading to T cell activation.

Response:

We thank reviewer for the comments. To further dissect the roles of PD-L1 on DC in T cell activation, cross-priming assay was performed ex vivo (**Figure 1e**). PD-L1-deficiency on DC significantly enhanced T cell priming and activation, which is consistent with our conclusions. The experiments in **Figure S2c-S2d (formerly S1ef)** have been repeated twice with similar results. We have revised the figure legend accordingly.

- In Figure 2b, an explanation should be given for the assessment of PD-L1^{high} expressing cells (%) as opposed to total PD-L1 expression.

Response:

As shown in **Fig 2a** and **3a-3c**, majority of cDC1s express a low level of PD-L1. Only a subset of cDC1s upregulate PD-L1 after tumor antigen uptake. Therefore, we compared the percentages of PD-L1^{high} cells in **Fig 2b**. We do have the total PD-L1 expression data (**Figure R4**). The trend of MFI (PD-L1) is similar to the percentage of PD-L1^{high} cells.

Figure R4. Related to figure 2b. MFIs of PD-L1 levels in cDC1 of naïve or MC38 tumor-bearing mice. Data are shown as mean ± SEM and are representative of two independent experiments. ***, p<0.001 determined by unpaired Student's t-test.

- In figure 2c more independent experiments plus quantification are needed of FTL-3 induced BMDC. Furthermore, the phenotype and vitality of generated cells should be provided to demonstrate robustness of BMDC cultures.

Response:

The quantification of PD-L1 levels in BMDC was shown (**Figure R5a**). The experiment has been independently repeated for three times. We have measured the surface expression of CD11c, MHC II, CD24, CD8a, and CD103 in the generated BMDC (**Figure R5b**). The staining patterns were consistent with previous study (PMID: 25100743).

Figure R5. Related to figure 2c. (a) Quantification of PD-L1 levels in FLT3-L-induced BMDC. Data are shown as mean \pm SEM. (b) Flow cytometry staining of CD11c, MHC II, CD24, CD8a, and CD103 in BMDC (Gray overlay: control). Data are representative of three independent experiments (a and b). FMO, fluorescence minus one; MFI, mean fluorescent intensities.

- In Figure 2 b+d, the expression level of PD-L1 is interrogated on cDC1. According the gating strategy in Sup. Fig 2 cDC1 are distinguished from cDC2 in these experiments. Could the expression levels of PD-L1 on cDC2 also be made available to accompany current figures?

Response:

Yes, we did stain PD-L1 on cDC2. The expression levels of PD-L1 on cDC2 were shown in **Figure 2b** and **2e** (formerly **2d**) in the revision.

- In Figure 2d, additional experiments with isolated cDC1 cells should be performed to investigate the direct involvement of IFN γ and IFN α /IFN β on PD-L1 expression on cDC1.

Response:

As suggested by the reviewer, we isolated DC and stimulated with both type I and II IFNs in vitro. We found that both type I and II IFNs were able to upregulate PD-L1 on cDC1 in vitro (**revised Figure 2d**). But only type II IFN was responsible for PD-L1 upregulation in vivo (**Figure 2e** (formerly **2d**)). We thank reviewer for the suggestions.

- In Figure 3a authors study tumor antigen uptake by cDC1 isolated from tumor and dLN. More repetitions plus quantification are needed to draw conclusions. Secondly, authors claim that cDC1 take up antigens in the tumor and transport these to the dLNs. This is overstated because the current

experimental setup does not allow drawing conclusions about the spatial and temporal dynamics of antigen uptake.

Response:

We thank reviewer for the comments. Quantification was shown in **Figure 3a**. The experiment has been repeated twice with similar results. And we agreed with the reviewer that our data was not sufficient to address antigen transportation. We have revised the text accordingly. We thank the reviewer for pointing it out.

• In Figure 3a authors illustrate that 1.95% of cDC1 in the dLN have taken up tumor antigen and proceed to show a histogram of the PD-L1 expression in this small fraction of cells in Fig 1b. Could a representative dot plot including absolute cell count of gated cells be shown for this small portion?

Response:

A representative dot plot with absolute cell counts and adjunct histograms were shown (**Figure R6**).

Figure R6. Related to figure 3b. Dot plot with absolute cell counts and adjunct histograms were shown. Data is representative of three independent experiments.

• In Figure 3c, authors nicely show that PD-L1 expression is increased upon tumor antigen uptake in cDC1 but not in cDC2. Is it possible to provide representative FACS plots of MC-38eGFP uptake by cDC2 similar to cDC1 in Fig3a?

Response:

We thank reviewer for the comments. Representative FACS plots of cDC2 were shown in the revised **Figure 3a**. We observed that cDC2 was able to uptake EGFP in tumor tissue. However, there was no/few EGFP+ cDC2 in dLN. These results were consistent with previous studies, which have shown that CD103+ migratory cDC1 were the major DC responsible for antigen cross-presentation in dLN (PMID: 27096321).

- Similarly in Fig 3d and e, an average of 2,5% of BMDCs take up tumor antigen of which the PD-L1 expression is shown. Here a representative and unadjusted dot plot should also be shown. Also more independent experiments plus quantification of PD-L1 expression on BMDCs are needed.

Response:

A representative dot plot were shown in **figure R7 (a-b)**. Quantification of PD-L1 expression was shown in **figure R7 (c)**. We found that PD-L1 level in EGFP+ BMDCs was higher than EGFP- BMDCs. However, such difference was not as significant as what we observed in vivo (**Figure 3b**). These data supports our conclusion that other factors were responsible for the further increase of PD-L1 in vivo (**Figure 3g**). Data shown are representative of three independent experiments.

Figure R7. (a) Related to figure 3d, representative dot plots were shown. (b-c) Related to figure 3e. (b) Dot plot with absolute cell counts and adjunct histograms were shown. (c) Quantification of PD-L1 levels in EGFP- and EGFP+ BMDC (n = 4). Data shown are representative of three independent experiments. **, p<0.01 determined by unpaired Student's t-test in (c).

- From Fig3fg, authors conclude that PD-L1 expression on cDC1 is up regulated by antigen uptake, IFN γ and CD8+ T cells. Additional insights and direct evidence should be provided to explain the dynamics or mechanisms underlying these sources of PD-L1 regulation on cDC1.

Response:

We thank reviewer for the suggestions. To further dissect the mechanisms of PD-L1 regulation, DC were isolated and co-cultured with activated T cells in vitro (**Figure 2f**). PD-L1 on DC was upregulated when co-cultured with activated T cells. The upregulation was reduced if IFN- γ was neutralized. This data was consistent with our study and strongly support our conclusions.

- In Figure 4abc, FTY720 treatment is used to block infiltration by T cells into early or more established tumors. Please, provide data demonstrating the extent of migration inhibition by T cells

in both early and late application of FTY720 treatment.

Response:

In the presence of FTY720, majority of the T cells disappeared from the peripheral blood (**Figure R8a**). In tumor tissues, the numbers of T cells also reduced, though not as significant as in peripheral blood (**Figure R8b**).

Figure R8. Efficacy of FTY720 treatment. Mice were treated with FTY720 as in **Figure 4a-4c**. T cell levels in PBMC (n = 5) (**a**) and tumor tissues (n = 4) (**b**) were shown. Data are shown as mean \pm SEM and are representative of three (**a**) or two (**b**) independent experiments. *, $p < 0.05$; **, $p < 0.01$ determined by unpaired Student's t-test in (**b**).

• In Fig 4d, authors claim that PD-L1 expression protects cDC1 from killing by cytotoxic T cells without any direct evidence. Additional experiments are required to support this conclusion.

Response:

To test the role of PD-L1 in T cell-mediated killing, we isolated DC, loaded with OT-1 peptides, and co-cultured them with activated OT-1 T cells for 4 hours at an E:T ratio of 3 (**Figure 4f**). The killing by cytotoxic T cells significantly increased in the absence of PD-L1 on DC. These were direct evidences showing that PD-L1 on DC protect them from killing by T cells. We thank reviewer for the suggestions.

REVIEWER COMMENTS

Reviewer #1 (Remarks to the Author):

The manuscript is much improved.

However an important point is missing as per my previous commentaries:

-Does PD-L1 deficiency in DC account for more potent endogenous antitumor immune responses?

This is not addressed by figure 1e.

This could be addressed now using for instance the EG7 model and the response to OVA. The prediction is that PD-L1 deficient mice in DC will have a more prominent expansion of SIINFEKL reactive CD8 T cells and perhaps CD4 T cells as well.

This question is very relevant for the overall message of this manuscript.

MHC tetramers or functional assays will provide an answer.

Reviewer #2 (Remarks to the Author):

Comments:

A new tumor model was introduced together with more repetitions and a greater variety of experiments. The text became clearer, reporting is more complete in the M&M section and a supplementary table of used reagents was provided as well. However the number of references should still be increased by addressing more research that is relevant to the current work and relate obtained results to more existing literature to build upon prior knowledge. Furthermore, I suggest that the revised additional text should also undergo professional editing as its clarity now contrasts with the improved original text.

Specific comments:

On page 7 "data not shown" is mentioned for GM-CSF induced BMDCs. Referring to non-provided data is to be avoided according the guidelines for authors and therefore the data should be provided as supplementary figure or the sentence should be removed.

On page 9 the EGFP+ MC38 model is introduced as method "to track antigen delivery". As previously agreed upon by the authors this is overstated and the description should be adjusted to model "to visualize antigen uptake".

In Figure R2 authors expand and further solidify the phenotype of isolated cDC1 and cDC2 used in experiments. Could these plots be added as supplementary data to the existing gating strategy? Regarding statistical testing, ANOVA's have been introduced where the means of multiple groups have been compared. However, in the M&M section under Statistical Analysis there is no mention of applied correction in case of multiple testing (of which there are numerous occasions).

In Figure R4 authors provide median mfi of PD-L1 expression on cDC1 in different tissues of naïve and tumor bearing mice. Could this figure be added as supplementary data as it provides additional insights into PD-L1 expression dynamics? It also offers the opportunity to relate displayed median mfi's of PD-L1 in Figure 2d, e, f to the median mfi's currently displayed in Figure R4.

In Figure R5a authors have displayed and quantified the positive PD-L1 expression level on FLT3-L-induced BMDCs. However, on page 7 in the text is mentioned that "FLT3-L-induced bone marrow-derived DC (BMDC) rarely expressed PD-L1" based on the not quantified data in figure 2c. The data of figure R5a should replace or be added to Figure 2c and the text should be adjusted accordingly. Could the authors also provide expression data of BMDCs treated with type I and type II interferons to support the conclusion of figure 2?

In Figure 2d, statistical testing is needed to determine significance of displayed differences.

In Figure R5b the phenotype of generated BMDCs is shown and validated. Could this data be included as supplementary figure as it supports all data obtained by BMDCs?

In the text there is no mention of the cDC2 data shown in Fig3c. Could the authors mention what the totality of cDC2 data in figure 3 implicates or at least how it should be interpreted?

In Figure R7c the increased expression of PD-L1 on BMDCs upon antigen uptake is quantified. This data should replace or be added to Figure 3e to support current conclusions.

In Figure 3g, in vivo studies show IFN γ dependent upregulation of PD-L1 during antigen uptake in cDC1. Could the authors also provide the data concerning EGFP negative CD1c PD-L1 expression during IFN γ /CD8 depletion? This will show us whether IFN γ /CD8 dependent upregulation of PD-L1 is restricted to cDC1s that have taken up antigen or is also the case without antigen-uptake.

In Figure R8, authors nicely show the effect of FTY720 treatment on T cell migration/circulation. Why did the authors choose to show CD3 as percentage of total viable cells instead of percentage of CD45+ cells, similar to Figure S2B and Figure S3B? Could the authors provide this data as supplementary figure as it supports the data displayed in figure 4?

Can the authors explain how migrating/tumor-infiltrating T cells are needed for effective anti-PD-L1 therapy in the early phase (according Figure 4a) but there is no significant difference in the frequency of tumor infiltrating T cells between Isotype and anti-PD-L1 treated mice in the early phase (according figure R8b)?

In Figure 4e, in the absence of T cells more CRE+ cDC2s die than CRE- cDC2s (Figure 4e) could the authors provide an explanation for this observation?

In Figure 4e, it becomes more apparent that PD-L1 expression reduces apoptosis in DC caused by the co-culture of activated T cells. However, the current data does not convincingly show that this apoptosis is induced via direct contact T cell mediated cytotoxicity towards the DCs upon antigen presentation, as is claimed by the authors.

Could the authors show that DCs loaded with irrelevant peptide co-cultured with activated OT-1 cells as well as DCs co-cultured with inactive T cells do not induce apoptosis in DCs?

Moreover, claims regarding the cell contact dependency of T cell – DC interaction leading to lowered DC viability should be proven using transwell setups of co-culture.

Reviewer #1 (Remarks to the Author):

The manuscript is much improved.

However an important point is missing as per my previous commentaries:

-Does PD-L1 deficiency in DC account for more potent endogenous antitumor immune responses?

This is not addressed by figure 1e.

This could be addressed now using for instance the EG7 model and the response to OVA. The prediction is that PD-L1 deficient mice in DC will have a more prominent expansion of SIINFEKL reactive CD8 T cells and perhaps CD4 T cells as well.

This question is very relevant for the overall message of this manuscript.

MHC tetramers or functional assays will provide an answer.

Response:

We thank reviewer for the comments and suggestions. To address this question, we inoculated DC-conditional PD-L1 knockout mice with E.G7 cells. After tumor established, OVA-reactive CD8+ T cells were measured by tetramer staining (**Figure 1f**). We observed that there were more OT-1-specific CD8+ T cells in conditional knockout mice. Together with the ELISPOT data in **Figure 1g** (formerly **1e**), these data suggest that PD-L1 deficiency in DC results in stronger endogenous antitumor immune responses.

Reviewer #2 (Remarks to the Author):

Comments:

A new tumor model was introduced together with more repetitions and a greater variety of experiments. The text became clearer, reporting is more complete in the M&M section and a supplementary table of used reagents was provided as well. However the number of references should still be increased by addressing more research that is relevant to the current work and relate obtained results to more existing literature to build upon prior knowledge. Furthermore, I suggest that the revised additional text should also undergo professional editing as its clarity now contrasts with the improved original text.

Response:

We thank reviewer for the comments and suggestions. We have addressed recent studies relevant to the current topic in the revised manuscript and cite more papers accordingly. We have had the additional text undergo professional editing as well.

Specific comments:

On page 7 “data not shown” is mentioned for GM-CSF induced BMDCs. Referring to non-provided data is to be avoided according the guidelines for authors and therefore the data should be provided as supplementary figure or the sentence should be removed.

Response:

The sentence has been removed as suggested.

On page 9 the EGFP+ MC38 model is introduced as method “to track antigen delivery”. As previously agreed upon by the authors this is overstated and the description should be adjusted to model “to visualize antigen uptake”.

Response:

The description has been adjusted.

In Figure R2 authors expand and further solidify the phenotype of isolated cDC1 and cDC2 used in experiments. Could these plots be added as supplementary data to the existing gating strategy?

Response:

These plots were added to the supplementary data as suggested (**Figure S3c**).

Regarding statistical testing, ANOVA's have been introduced where the means of multiple groups have been compared. However, in the M&M section under Statistical Analysis there is no mention of applied correction in case of multiple testing (of which there are numerous occasions).

Response:

Information regarding the applied correction was added to the M&M section.

In Figure R4 authors provide median mfi of PD-L1 expression on cDC1 in different tissues of naïve and tumor bearing mice. Could this figure be added as supplementary data as it provides additional insights into PD-L1 expression dynamics? It also offers the opportunity to relate displayed median mfi's of PD-L1 in Figure 2d, e, f to the median mfi's currently displayed in Figure R4.

Response:

This data was added to the supplementary data (**Figure S3d**) as suggested.

In Figure R5a authors have displayed and quantified the positive PD-L1 expression level on FLT3-L-induced BMDCs. However, on page 7 in the text is mentioned that “FLT3-L-induced bone marrow-derived DC (BMDC) rarely expressed PD-L1” based on the not quantified data in figure 2c. The data of figure R5a should replace or be added to Figure 2c and the text should be adjusted accordingly. Could the authors also provide expression data of BMDCs treated with type I and type II interferons to support the conclusion of figure 2?

Response:

Figure R5a was added to **Figure 3c** (formerly **2c**). The text was adjusted accordingly.

As suggested by the reviewer, we treated BMDCs with type I and II IFN (**Figure 3d**). PD-L1 was dramatically upregulated, similar to what was observed in purified primary DCs (**Figure 3e**).

In Figure 2d, statistical testing is needed to determine significance of displayed differences.

Response:

Statistical analysis was added to **Figure 3e** (formerly **2d**).

In Figure R5b the phenotype of generated BMDCs is shown and validated. Could this data be included as supplementary figure as it supports all data obtained by BMDCs?

Response:

This data was added to the supplementary data as suggested (**Figure S4**).

In the text there is no mention of the cDC2 data shown in Fig3c. Could the authors mention what the totality of cDC2 data in figure 3 implicates or at least how it should be interpreted?

Response:

The interpretation to cDC2 data was added to the text. (Page 8, “No significant difference was observed in cDC2s, suggesting that mechanisms regulating PD-L1 expression in these cells

might be different.”)

In Figure R7c the increased expression of PD-L1 on BMDCs upon antigen uptake is quantified. This data should replace or be added to Figure 3e to support current conclusions.

Response:

Figure R7c was added to **Figure 4e** (formerly **3e**).

In Figure 3g, in vivo studies show IFN γ dependent upregulation of PD-L1 during antigen uptake in cDC1. Could the authors also provide the data concerning EGFP negative CD1c PD-L1 expression during IFN γ /CD8 depletion? This will show us whether IFN γ /CD8 dependent upregulation of PD-L1 is restricted to cDC1s that have taken up antigen or is also the case without antigen-uptake.

Response:

The data of EGFP-negative cDC1 was shown in **Figure 4g** (formerly **3g**).

In Figure R8, authors nicely show the effect of FTY720 treatment on T cell migration/circulation. Why did the authors choose to show CD3 as percentage of total viable cells instead of percentage of CD45+ cells, similar to Figure S2B and Figure S3B? Could the authors provide this data as supplementary figure as it supports the data displayed in figure 4?

Response:

After FTY720 treatment, some tumors sizes were larger than others. Therefore, we thought it would be better to compare the absolute T cell numbers. However, we did have the data of CD3 as percentage of CD45+ cells, which was provided in **Figure S5b**.

Can the authors explain how migrating/tumor-infiltrating T cells are needed for effective anti-PD-L1 therapy in the early phase (according Figure 4a) but there is no significant difference in the frequency of tumor infiltrating T cells between Isotype and anti-PD-L1 treated mice in the early phase (according figure R8b)?

Response:

This is a good question! In fact, many T cells inside tumor tissues might not be tumor specific. During early phase, more priming occurs inside draining LN, most of them reach to tumor tissues on day 7-12 after inoculation. Therefore, ongoing migrating antigen specific T cells could be greatly reduced at early phase by FTY720 treatment while total number of T cells might not be changed. Increased priming on draining LN might also occur when tumors are irradiated or under chemotherapy and the role of migrating T cells might become more important (PMID: 31477729, 30209192). Once most antigen-specific T cells have arrived tumor tissues at late phase (after day 14), they become essential to control tumor while smaller fraction of T cells in draining LN plays minor role. Data in **Figure R8b** is one representative of two repeats (n = 4).

The data showed a moderate increase of tumor infiltrating T cells (TILs) after anti-PD-L1 treatment. The data pooling two repeats (n = 8) was shown in **Figure R9a** and **R9b** (the latter was provided as **Figure S5b**). The difference between IgG and anti-PD-L1 (day 8, without FTY720) was statistical significant as compared by unpaired Student's t test (p=0.0126 in **Figure R9a**, p=0.0192 in **Figure R9b**).

Figure R9. Efficacy of FTY720 treatment. Mice were treated with FTY720 as in **Figure 5a-5c**. T cell levels in tumor tissues (n = 8) were shown as percentage of total live cells (**a**) or CD45+ cells (**b**). Data are shown as mean ± SEM and are pool of two independent experiments. n.s., not significant; *, p<0.05; **, p<0.01; ***, p<0.001 determined by unpaired Student's t-test.

In **Figure 4e**, in the absence of T cells more CRE+ cDC2s die than CRE- cDC2s (**Figure 4e**) could the authors provide an explanation for this observation?

Response:

In this experiment, DCs were isolated from mice, loaded with peptide, and cultured for 4 hours before flow cytometry analysis. Thus, the data in this group mainly resembled their viability in vivo. We did observe that there were some differences in the viability of DCs from Cre- and Cre+ mice (**Figure 5d**) (formerly **4d**). And the difference was more significant in cDC2 comparing to cDC1. There are two possible explanations: 1> Different contributions of PD-L1 to survival due to intrinsic differences of the cells; and/or 2> cDC2 has higher endogenous PD-L1 expression (**Figure 3b**). However, the answer to this question is outside the scope of the current study.

In **Figure 4e**, it becomes more apparent that PD-L1 expression reduces apoptosis in DC caused by the co-culture of activated T cells. However, the current data does not convincingly show that this apoptosis is induced via direct contact T cell mediated cytotoxicity towards the DCs upon antigen presentation, as is claimed by the authors.

Could the authors show that DCs loaded with irrelevant peptide co-cultured with activated OT-1 cells as well as DCs co-cultured with inactive T cells do not induce apoptosis in DCs?

Moreover, claims regarding the cell contact dependency of T cell – DC interaction leading to lowered DC viability should be proven using transwell setups of co-culture.

Response:

As suggested by the reviewer, we loaded DCs with OT-1 or SIY peptide and co-cultured them with activated or inactive OT-1 T cells. Viability of DCs loaded with OT-1 peptide significantly reduced in the presence of activated OT-1 T cells (**Figure R10a**, grey vs red). Cell death in DCs loaded with SIY peptide is not as high as DCs loaded with OT-1 peptide (**Figure R10a**, blue vs red), suggesting that antigen-specificity is important. However, some cells still died in DCs loaded with SIY (**Figure R10a**, grey vs blue). This might be a result of off-target killing from activated T cells, or other antigen-independent mechanisms, which is consistent with previous studies (PMID: 29507197). When DC were co-cultured with inactive T cells, limited DCs were killed, suggesting that the killing required T cell activity (**Figure R10a**, grey vs green).

No significant cell death was observed in a transwell assay, suggesting that majority of these effects required cell-cell contact (**Figure R10b**). This data was provided as **Figure S6**.

Figure R10. Characterization of T cell-mediated killing. (a) Isolated DCs were loaded with OT-1 or SIY peptide and incubated with activated or inactive OT-1 T cells at an E:T ratio of 3 for 4 hours. Cell death of DCs was measured by flow cytometry (n = 3). (b) Isolated DCs were loaded with OT-1 peptide. Activated OT-1 T cells were seeded on the upper side of a transwell membrane, while DCs were seeded on the lower side. Cell death was measured 4 hours later (n = 3 or 4). Data are shown as mean \pm SEM and are representative of two independent experiments. n.s., not significant; **, $p < 0.01$; ***, $p < 0.001$ determined by unpaired Student's t-test.

REVIEWERS' COMMENTS:

Reviewer #1 (Remarks to the Author):

My comment has been addressed

Reviewer #2 (Remarks to the Author):

Comments:

Additional (control) experiments have been performed and complementary datasets have been provided to further solidify the author's claims. The discussion was extended to place current work in context of relevant studies. Overall the manuscript has greatly improved, especially considering the first version. Some minor comments regarding clarity remain.

Specific comments:

In Figure S3(d and e) the spelling of 'naïve' is incorrect.

On page 8 it states 'cDC2s uptook antigens' which should be 'cDC2s took up antigens' or 'cDC2s have taken up antigens'

As multiple tumor models are now introduced, a suggestion to improve clarity is to specify which tumor model is used when stating 'tumor-bearing' in a figure legend. Especially in regard to Figure 3b and Fig S4e.

The manuscript might benefit of an illustrated schematic depicting the proposed role(s) of PD-L1 on DCs in context of anti-tumor immunity during antigen uptake, T cell priming and T cell – DC cytotoxicity and how this is affected by anti-PD-L1 blockade therapy.

Reviewer #1 (Remarks to the Author):

My comment has been addressed

Reviewer #2 (Remarks to the Author):

Comments:

Additional (control) experiments have been performed and complementary datasets have been provided to further solidify the author's claims. The discussion was extended to place current work in context of relevant studies. Overall the manuscript has greatly improved, especially considering the first version. Some minor comments regarding clarity remain.

Specific comments:

In Figure S3(d and e) the spelling of 'naïve' is incorrect.

Response:

We are sorry for the typo. The spelling has been corrected.

On page 8 it states 'cDC2s uptook antigens' which should be 'cDC2s took up antigens' or 'cDC2s have taken up antigens'

Response:

As suggested by the reviewer, the sentence was revised to "cDC2s took up antigens".

As multiple tumor models are now introduced, a suggestion to improve clarity is to specify which tumor model is used when stating 'tumor-bearing' in a figure legend. Especially in regard to Figure 3b and Fig S4e.

Response:

Tumor models have been specified in all figure legends.

The manuscript might benefit of an illustrated schematic depicting the proposed role(s) of PD-L1 on DCs in context of anti-tumor immunity during antigen uptake, T cell priming and T cell – DC cytotoxicity and how this is affected by anti-PD-L1 blockade therapy.